# Prevalence and Prognostic Role of IDH Mutations in Acute Myeloid Leukemia: Results of the GIMEMA AML1516 Protocol

**DOI:** 10.3390/cancers14123012

**Published:** 2022-06-18

**Authors:** Monica Messina, Alfonso Piciocchi, Tiziana Ottone, Stefania Paolini, Cristina Papayannidis, Federica Lessi, Nicola Stefano Fracchiolla, Fabio Forghieri, Anna Candoni, Andrea Mengarelli, Maria Paola Martelli, Adriano Venditti, Angelo Michele Carella, Francesco Albano, Valentina Mancini, Bernardi Massimo, Valentina Arena, Valeria Sargentini, Mariarita Sciumè, Domenico Pastore, Elisabetta Todisco, Giovanni Roti, Sergio Siragusa, Marco Ladetto, Stefano Pravato, Eleonora De Bellis, Giorgia Simonetti, Giovanni Marconi, Claudio Cerchione, Paola Fazi, Marco Vignetti, Sergio Amadori, Giovanni Martinelli, Maria Teresa Voso

**Affiliations:** 1GIMEMA Foundation, 00182 Roma, Italy; m.messina@gimema.it (M.M.); a.piciocchi@gimema.it (A.P.); v.arena@gimema.it (V.A.); v.sargentini@gimema.it (V.S.); p.fazi@gimema.it (P.F.); m.vignetti@gimema.it (M.V.); 2Ematologia, Dipartimento di Biomedicina e Prevenzione, Università di Roma Tor Vergata, 00133 Roma, Italy; tiziana.ottone@uniroma2.it (T.O.); adriano.venditti@uniroma2.it (A.V.); sergio.amadori1946@gmail.com (S.A.); 3Neuro-Oncohematology Unit, IRCCS Fondazione Santa Lucia, 00179 Roma, Italy; 4IRCCS Azienda Ospedaliero-Universitaria di Bologna Istituto di Ematologia “Seràgnoli” Bologna, 40138 Bologna, Italy; stefania.paolini@unibo.it (S.P.); cristina.papayannidis@unibo.it (C.P.); 5Ematologia ed Immunologia Clinica, Università degli Studi di Padova, 1222 Padua, Italy; lessi.federica@gmail.com (F.L.); stefano.pravato@aopd.veneto.it (S.P.); 6UOC Ematologia, Fondazione IRCCS Ca’ Granda Ospedale Maggiore Policlinico, 20122 Milano, Italy; nicola.fracchiolla@policlinico.mi.it (N.S.F.); mariarita.sciume@policlinico.mi.it (M.S.); 7UO Ematologia-AOU Policlinico di Modena, 41125 Modena, Italy; fabio.forghieri@unimore.it; 8Clinica Ematologica, ASUFC, Università degli Studi di Udine, 33100 Udine, Italy; candoni.anna@aoud.sanita.fvg.it; 9UO Ematologia-IRCCS Istituto Nazionale Tumori Tumori Regina Elena, 00128 Roma, Italy; andrea.mengarelli@ifo.it; 10Sezione di Ematologia ed Immunologia Clinica, Università degli Studi di Perugia, 06123 Perugia, Italy; maria.martelli@unipg.it; 11Ematologia e Centro Trapianti CSE Fondazione IRCCS Casa Sollievo della Sofferenza, 71013 San Giovanni Rotondo, Italy; am.carella@operapadrepio.it; 12Hematology and Stem Cell Transplantation Unit, Department of Emergency and Organ Transplantation (D.E.T.O.), University of Bari Aldo Moro, 70121 Bari, Italy; francesco.albano@uniba.it; 13Ospedale Niguarda Ca Granda-SC Ematologia Blocco SUD, 20162 Milano, Italy; valentina.mancini@ospedaleniguarda.it; 14IRCCS San Raffaele Scientific Institute, 20132 Milano, Italy; bernardi.massimo@hsr.it; 15UOC Ematologia Brindisi, 72100 Brindisi, Italy; domenico.pastore0@gmail.com; 16Onco-Hematology Division, IEO European Institute of Oncology IRCCS, 20141 Milan, Italy; elisabetta.todisco@asst-valleolona.it; 17Azienda Ospedaliera Universitaria di Parma, Ematologia, Università di Parma, 43126 Parma, Italy; giovanni.roti@unipr.it; 18U.O. di Ematologia con Trapianto-A.U. Policlinico Paolo Giaccone, 90127 Palermo, Italy; sergio.siragusa@unipa.it; 19AO SS Antonio e Biagio Arrigo, 15121 Alessandria, Italy; marco.ladetto@ospedale.al.it; 20Hematology Unit, Azienda Sanitaria Universitaria Giuliano Isontina, 34148 Trieste, Italy; debellis.eleonora.1@gmail.com; 21Biosciences Laboratory, IRCCS Istituto Romagnolo per lo Studio dei Tumori (IRST) Dino Amadori, 47014 Meldola, Italy; giorgia.simonetti@irst.emr.it (G.S.); giovanni.martinelli@irst.emr.it (G.M.); 22Hematology Unit, Istituto Scientifico Romagnolo per lo Studio e la Cura dei Tumori [M1] (IRST) IRCCS, 47014 Meldola, Italy; giovanni.marconi@irst.emr.it (G.M.); claudio.cerchione@irst.emr.it (C.C.)

**Keywords:** AML, DH1, IDH2, prevalence, prognosis

## Abstract

**Simple Summary:**

*IDH1/2* mutations are a common event in acute myeloid leukemia (AML) and represent a therapeutic target. We designed the GIMEMA AML1516 observational protocol to examine the prevalence of *IDH1/2* mutations and the associations between IDH mutations and clinico-biological parameters in a cohort of Italian patients affected by AML. By analyzing 284 consecutive adult AML patients, we confirmed that *IDH1* and *IDH2* mutations are frequently detected–14% and 18%, respectively–at diagnosis. *IDH1/2* mutations were significantly associated with an inferior performance status and non-complex karyotype when compared to *IDH1/2*-WT. With regards to the outcome, in the subset of *IDH1/2*-mutated patients the rate of complete remission achievement was 60.5% and overall survival at 2 years was 44.5%: these percentages did not significantly differ from *IDH1/2*-WT patients. However, given the availability of *IDH1/2* inhibitors, it is important to recognize IDH1/2-mutated cases up-front to offer patients the most appropriate therapeutic strategy.

**Abstract:**

*IDH1*/*2* mutations are common in acute myeloid leukemia (AML) and represent a therapeutic target. The GIMEMA AML1516 observational protocol was designed to study the prevalence of *IDH1*/*2* mutations and associations with clinico-biological parameters in a cohort of Italian AML patients. We analyzed a cohort of 284 AML consecutive patients at diagnosis, 139 females and 145 males, of a median age of 65 years (range: 19–86). Of these, 38 (14%) harbored *IDH1* and 51 (18%) *IDH2* mutations. *IDH1/2* mutations were significantly associated with WHO PS >2 (*p* < 0.001) and non-complex karyotype (*p* = 0.021) when compared to *IDH1/2*-WT. Furthermore, patients with *IDH1* mutations were more frequently *NPM1*-mutated (*p* = 0.007) and had a higher platelet count (*p* = 0.036). At relapse, *IDH1*/2 mutations were detected in 6 (25%) patients. As per the outcome, 60.5% of *IDH1/2*-mutated patients achieved complete remission; overall survival and event-free survival at 2 years were 44.5% and 36.1%, respectively: these rates were similar to *IDH1*/*2*-WT. In *IDH1*/*2*-mutated patients, high WBC proved to be an independent prognostic factor for survival. In conclusion, the GIMEMA AML1516 confirms that *IDH1*/*2* mutations are frequently detected at diagnosis and underlines the importance of recognizing *IDH1*/*2*-mutated cases up-front to offer the most appropriate therapeutic strategy, given the availability of IDH1/2 inhibitors.

## 1. Introduction

Progresses in the knowledge of the genetic landscape of AML—accelerated by high throughput sequencing technologies—led to a better understanding of AML pathogenesis and enhanced the development of targeted approaches.

Mutations targeting epigenetic regulators emerged as one of the most common events—accounting for >50% of AML patients—and contribute to the differentiation block typical of AML [1,2]. Isocitrate dehydrogenase 1 and 2 (IDH1 and IDH2) belong to the class of epigenetic modulators and mutations of these genes occur in up to 20% of adult AML cases [3,4,5,6] and 30% of pediatric AML [7]. *IDH1* and *IDH2* mutations target the conserved arginine residues, namely R132 of IDH1, and R140 and R172 of IDH2 [8]. They determine an aberrant production of 2-hydroxyglutarate (2HG) that acts as an antagonist of α-KG; thus, inhibiting the activity of multiple α-KG-dependent dioxygenases, including both histones and DNA demethylases involved in epigenetic control of gene expression. As a consequence, *IDH1* and *IDH2* mutations determine an aberrant hypermethylated phenotype and, ultimately, influence cell differentiation [4,9].

*IDH1/2* mutations are associated with intermediate-risk cytogenetics and *NPM1* mutations [4,10], in particular, in the absence of DNA-damage-related and cohesin gene mutations [11]. Moreover, Chou et al. reported an association between *IDH1*/*2* mutations and higher platelet counts, normal karyotype, and isolated trisomy 8 [12,13,14].

The impact of *IDH1/2* mutations on AML prognosis is controversial, and depends on the specific AML subsets, i.e., normal karyotype or *FLT3*-WT/*NPM1*-WT, or treatment groups, i.e., standard intensive chemotherapy [5,6,15,16]. In other cohorts, *IDH1*/*2* mutations do not impact on prognosis [17,18].

Despite this, IDH1 and IDH2 immediately qualified as promising therapeutic targets, due to the activating nature of their recurrent mutations: small inhibitory molecules have been developed and were tested in clinical trials, as monotherapy or in combination with chemotherapy or azacitidine [19,20,21,22]. Two orally bioavailable IDH inhibitors, enasidenib (IDH2 inhibitor) and ivosidenib (IDH1 inhibitor) are now FDA-approved [19,20]: the former for the treatment of *IDH2*-mutated relapsed-refractory AML, and the latter for both *IDH1*-mutated relapsed-refractory and newly diagnosed AML unfit for intensive chemotherapy.

This progress has a substantial impact on the therapeutic algorithm of AML patients harboring *IDH1/2* mutations. Therefore, the assessment of *IDH1/2* mutations is pivotal to identify the population of patients that might benefit from the use of IDH inhibitors, at diagnosis or at relapse.

We present here the results of the GIMEMA AML1516 protocol, designed to (i) study the prevalence of *IDH1* and *IDH2* mutations in patients with AML at the time of initial diagnosis and at relapse, (ii) evaluate the association between *IDH* mutations and patient or disease characteristics, and (iii) assess the impact on response to treatment and survival.

## 2. Materials and Methods

### 2.1. GIMEMA AML1516 Study Design

The GIMEMA AML1516 protocol (ClinicalTrials.gov Identifier: NCT02986620) is an observational study aimed at collecting data on *IDH1* and *IDH2* mutational status in adult AML patients in Italy treated as per clinical practice, not including IDH1/2 inhibitors. The primary objective of the trial was to estimate the prevalence and type of *IDH* mutations in AML at initial diagnosis and relapse. The secondary objectives were to evaluate the associations between *IDH* mutations and clinico-biological parameters (i.e., age, white blood cell (WBC), lactate dehydrogenase (LDH), cytogenetics, *NPM1*, *FLT3*-ITD, *CEBPA* alterations), AML type, treatment response, and survival.

The study was active starting from May 2017 to January 2020 and included a retrospective and a prospective cohort.

The analysis of *IDH1* and *IDH2* mutations was performed either by Sanger sequencing or NGS technologies at local laboratories [8,23]. The sensitivity of Sanger sequencing analysis is approximately 15–20% and the presence of chromatograms with a double peak into wild-type gene sequence identified an *IDHs* gene mutation. For the assessment of *IDHs* status by NGS assay, the detection limit of the variant allele frequency (VAF) was 5%.

Study data were collected and managed using REDCap electronic data capture tools hosted at GIMEMA Foundation [24,25].

### 2.2. Statistical Analysis

Characteristics of patients were summarized by means of cross-tabulations or quantiles.

*IDH1*-*IDH2* mutation detection was evaluated in terms of percentage of patients at the time of initial diagnosis and relapse.

Non-parametric tests were applied, in univariate analysis, for comparisons between groups, chi-squared and Fisher exact test for difference in terms of categorical variables or mutation rate, Mann–Whitney and Kruskal–Wallis tests for difference in terms of continuous variables. All clinical parameters, genetic subtypes, and treatment received were considered in the univariate analyses. The multivariate models considered all relevant clinical/biologic variables or covariates with a *p*-value less than 0.15 in the univariate analysis.

Logistic regression models were used in univariate and multivariate analyses to assess if the clinical and biological parameters are associated to response outcomes (CR and ORR rate). Odds ratios (OR) and 95% confidence intervals were reported as parameter results of the logistic regression models.

Survival distributions (e.g., overall survival (OS), event-free survival (EFS)) were estimated using the Kaplan–Meier product limit estimator. Subgroup comparisons with clinical and biological parameters were performed for descriptive purposes.

Differences in terms of time to response OS and EFS were evaluated by means of log-rank test or Cox regression model in univariate and multivariate analyses, after assessment of proportionality of hazards.

Hazard ratios (HR) and 95% confidence interval were reported as parameter results of the Cox regression models.

## 3. Results

### 3.1. Study Population

Between 5/2017 and 1/2020, 393 consecutive patients were diagnosed with AML at 17 Italian Hematology Centers and members of the GIMEMA working group, enrolled in the AML1516 study. Of them, 388 were deemed eligible. *IDH1/2* mutational status was available for 361 patients (337 at diagnosis and 24 at relapse).

The present analysis is based on 284 patients studied at diagnosis with available *IDH1/2* mutation status, treatment and follow-up data. At diagnosis, 145 (51%) patients were males and 139 (49%) were females. Median age was 65 (range 19–86) years. In total, 229 (81%) patients had a de novo, 37 (13%) a secondary, and 16 (5.7%) a therapy-related AML. Cytogenetics was available for 259 patients, 132 (50.9%) had a normal and 29 (11.2%) a complex karyotype. As per the main chromosomal aberrations, anomalies of chromosome 5 (del5q, monosomy 5) occurred in 20 patients (7.7%), and aberrations of chromosome 7 (del7q, monosomy 7) were detected in a total of 20 patients (7.7%); in 23 patients a trisomy 8 was documented. Recurrent rearrangements, including *RUNX1T1-RUNX1* and *CBF-MYH11* were detected in 5 (1.9%) and 10 (3.9%) patients. *FLT3* mutations were detected in 60/271 (22.4%), *NPM1* in 71/266 (26.7%). 

With regards to the treatment received, 201 (71%) patients were treated with conventional chemotherapy, 76 (27%) with hypomethylating agents, and the remaining 7 patients with other treatment schemes.

Demographic characteristics are summarized in Table 1.

### 3.2. Incidence, Type of IDH1/2 Mutations, and Patients’ Clinico-Biological Features

Of 284 patients studied at diagnosis, 38 (14%) carried *IDH1* mutations and 51 (18%) *IDH2* mutations (Figure 1A). With regards to the type of *IDH1* mutations, the majority (32, 84.2%) targeted R132, with R132C and R132H being the most common substitutions. Similarly, R140 was the most commonly involved residue of *IDH2* (30, 58.8%)—with R140Q accounting for the vast majority of substitutions—followed by R172K detected in 19 cases (37.2%), as depicted in Figure 1B,C.

*IDH1/2* mutations were significantly associated with WHO PS >2 (*p* < 0.001) and non-complex karyotype (*p* = 0.021) when compared to *IDH1/2*-WT. As per MDS-related anomalies, a WT status of IDH1/IDH2 was associated with del5q (*p* = 0.035). Furthermore, patients with *IDH1* mutations had higher platelet counts (*p* = 0.036) and were more frequently *NPM1*-mutated (*p* = 0.007, Table 1).

At relapse, 5 (21%) patients carried *IDH1* mutations, all targeting R132 with R132H being the most common substitution (3 out of 5); 1 patient had a concomitant *FLT3*-ITD mutation and another patient a concurrent *TP53* mutation. Only 1 patient (4.2%) harbored a *IDH2* mutation, that targeted the R172 residue.

### 3.3. Treatment Response and Survival According to IDH1/2 Mutations

Out of 284 patients with therapy information, 201 (71%) were treated with a conventional chemotherapy approach (CHT), 76 (27%) with a hypomethylating agent (HMA), and 7 (2.5%) with other regimens.

Overall, 228 patients were evaluable for response and 128 (56%) achieved complete remission (CR). When considering CHT vs. HMA, 113 of 181 (62.4%) treated with conventional CHT achieved a CR while only 13 of 41 (31.7%) treated with HMA obtained a CR (*p* < 0.0001). There were no differences in CR rate when stratifying patients according to *IDH1/2* mutational status (60.5% CR in IDH1/2-mutated vs. 64% in IDH1/2-WT patients).

Indeed, the parameters with an impact on CR, resulting from the logistic regression model, were younger age (OR 0.96 95% CI 0.94–0.98, *p* < 0.001), WHO performance status (OR 0.2 95% CI 0.07, 0.51, *p* = 0.001), de novo AML (OR 0.17, 95% CI 0.06–0.39, *p* < 0.001), *NPM1* mutations (OR 2.26, 95% CI 1.29–4.42, *p* = 0.013), and conventional CHT treatment (OR 0.28, 95% CI 0.13–0.57, *p* < 0.001, Appendix A). WHO PS and AML type retained statistical significance also in the multivariate model.

Overall response was obtained by 167 (73%) patients, at a similar rate in IDH1/2-mutated (71%) and *IDH1*/*2*-WT (68%) patients.

With a median follow-up of 22.5 months (13.5–35.7), overall survival (OS) at 24 months was 43.7% (95% CI 37.5–50.9) and event-free survival (EFS) was 30% (95% CI 24.5–36.6).

Overall, there were no differences in OS or EFS in patients with *IDH1*/*2*-mutated vs. WT AML (44.5% vs. 43.3% Figure 2A, and 36.1% vs. 26.6%, Figure 2B, respectively).

Additionally, we did not document any difference in OS and EFS grouping patients according to the *IDH* mutation type (IDH1-R132 vs. IDH2-R140 vs. IDH2-R172 vs. IDH-WT, Figure 2C,D).

Analyzing the clinico-biological parameters that impact on survival outcomes, the univariate analyses showed that patients treated with CHT when compared to HMT, had a significantly longer OS (49.5% vs. 21%, *p* < 0.001) and EFS (34.4% vs. 15.3%, *p* = 0.0013). Furthermore, in the univariate model, age (HR 1.04, 95% CI 1.02–1.05, *p* < 0.001), high WBC (HR 1.0, 95% CI 1.0–1.0, *p* = 0.004), WHO PS (1 vs. 0: HR 1.9, 95% CI 1.3–2.9, *p* < 0.0001; 2 vs. 0: HR 2.7, 95% CI 1.6–4.5, *p* < 0.0001), complex karyotype (HR 2.83, 95% CI 1.79–4.45, *p* < 0.001), and HMA treatment (HR 2.2, 95% CI 1.56–3.10, *p* < 0.001) negatively impacted on OS and EFS. Age, WHO PS, and complex karyotype proved independent prognostic factors for OS and EFS in the multivariable model, as detailed in Table 2.

When restricting the analysis to *IDH1*/*2*-mutated patients, the univariate analyses for OS and EFS confirmed also in this AML subset that age (HR 1.03, 95% CI 1.00–1.06, *p* = 0.019), WBC (HR 1.0, 95% CI 1.00–1.00, *p* = 0.007), and HMA (HR 2.03, 95% CI 1.07–3.83, *p* = 0.030) were associated with inferior survival. In the multivariate analysis, only WBC was confirmed as an independent prognostic factor (Table 2).

## 4. Discussion

The GIMEMA AML1516 study illustrates the prevalence of *IDH1/2* mutations in AML in Italy and adds further evidence on the value of screening these aberrations for therapeutic purposes.

By analyzing a cohort of 284 adult AML mainly de novo, we found that 32% of patients carried either a *IDH1* or *IDH2* mutation at diagnosis. Compared to the literature, this incidence is higher than reported in other patient cohorts (30% vs. 20%) and this may be due to the observational nature of the study, and to a selection bias [3,4,5,6]. As already reported, the most common IDH1 changes were R132C and R132H, while R140Q and R172K were the most frequent substitutions in IDH2 [8]. As per the association with clinico-biological parameters, we documented that *IDH1*/*2*-mutated AML had more frequently a WHO PS >2. We did not document any association with a specific cytogenetic subset, but we found that *IDH1*/*2*-mutated AML more frequently display a non-complex karyotype. Additionally, *IDH1* mutation was associated with higher platelet counts and with *NPM1* mutations, in agreement with Chou et al. [12].

Next, we analyzed the outcome of patients with *IDH1*/*2*-mutated AML in comparison with *IDH1*/*2*-WT. We did not document any difference neither in terms of CR achievement, nor of survival. In our cohort, the CR rate was 60% in *IDH1*/*2*-mutated cases, while OS and EFS at 24 months were 44% and 36%, respectively. Additionally, there was no impact on outcome when grouping patients according to common *IDH1*/*2* mutations (i.e., IDH1-R132, IDH2-R140, and IDH2-R172).

Therefore, our data reinforce the notion that *IDH1*/*2* mutational status does not impact on prognosis, in line with the reports by Di Nardo and Chotirat [17,18]. Accordingly, Middeke et al. recently reported on the prognostic role of *IDH1*/*2* mutations in the largest AML cohort treated with intensive chemotherapy [26]. They did not find any difference in response rates, nor in survival for patients carrying *IDH1*/*2* mutations when compared to *IDH1*/*2*-WT patients. However, when the most common IDH1/2 substitutions were analyzed, they found that IDH1-R132C is associated with a lower rate of complete remission and a trend towards shorter OS compared to other *IDH1* mutations and *IDH1*/*2*-WT. On the contrary, patients with IDH2-R172K-mutated AML had a better OS within the ELN2017 intermediate/adverse risk groups, compared to *IDH1*/*2*-WT. Most likely, we failed to detect this difference because of the low number of cases included in these subgroups.

Other parameters, such as age and WBC count, had an impact on survival, independent of *IDH* mutations.

Lastly, we evaluated the incidence of *IDH1*/*2* mutations at relapse and we found that 25% of cases were *IDH1*/*2* mutated. However, this data relies on the analysis of only 24 patients; thus, representing a limitation of the present study. To this regard, Chou and colleagues reported that *IDH* mutations are stable during disease course and even detected at CR [12,27].

As discussed above, the prognostic relevance of *IDH1/2* mutations is not straightforward. Notwithstanding, IDH1 and IDH2 display a prominent therapeutic role since they can be pharmacologically targeted. In the past few years, clinical trials showed that IDH1/2 inhibitors are well-tolerated and efficacious as monotherapy. In particular, the first IDH inhibitors approved by FDA were enasidenib, a selective allosteric inhibitor of IDH2-mutated, and ivosidenib that competes with magnesium for binding to mutated IDH1 enzyme.

Both were approved for the treatment of relapsed/refractory AML and induced a CR/CRh rate of approximately 30% [19,20], and a median OS of roughly 9 months. Subsequently, ivosidenib was approved also for newly diagnosed patients ineligible for chemotherapy. In this subset, the CR/CRh rate was 42% and OS 12% [28]. In newly diagnosed AML, further improvements were obtained with the use of combination approaches. Indeed, in combination with azacytidine and with the standard 7 + 3 intensive chemotherapy approach, IDH1/2 inhibitors induced CR/CRh rates exceeding 60% [21,22,29,30,31]. Another option is the combination with venetoclax that is highly effective in *NPM1*/*IDH*-mutated AML [32].

Second generation and “pan” IDH inhibitors are currently under development.

This progress broadens the therapeutic armamentarium of *IDH1*/*2*-mutated AML; thus, contributing to the shift to targeted regimens alone or in combination or in sequence. Therefore, the possible use of these strategies makes the screening of *IDH1*/*2* mutations of utmost importance.

## 5. Conclusions

The GIMEMA AML1516 confirms that *IDH1/2* mutations are frequently detected at diagnosis in an Italian AML cohort of patients and underlines the importance of recognizing *IDH1*/*2*-mutated cases up-front to offer patients the most appropriate therapeutic strategy.

## Figures and Tables

**Figure 1 cancers-14-03012-f001:**
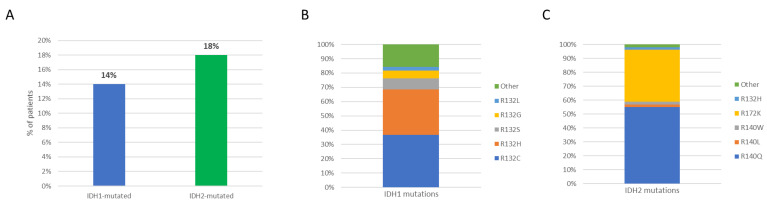
(**A**) Incidence of *IDH1*/*2* mutations at AML diagnosis; distribution of *IDH1* (**B**) and *IDH2* (**C**) mutation subtypes.

**Figure 2 cancers-14-03012-f002:**
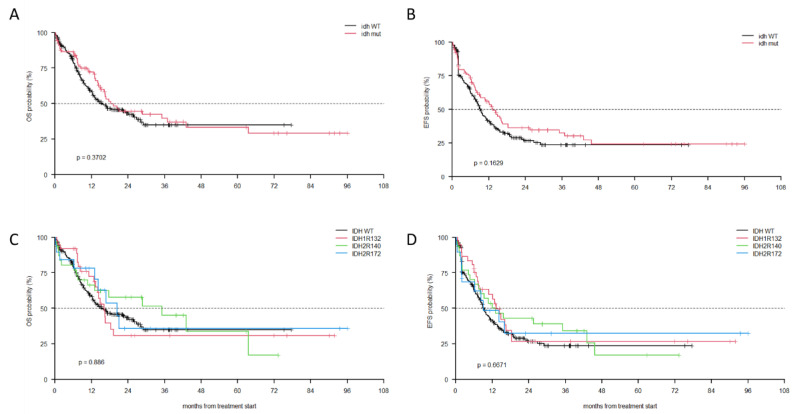
(**A**) OS by *IDH1/2* mutations; (**B**) EFS by *IDH1/2* mutations; (**C**) OS by the most frequent *IDH1/2* mutations; (**D**) EFS by the most frequent *IDH1/2* mutations.

**Table 1 cancers-14-03012-t001:** Patient characteristics by *IDH1*/*2* mutations.

		*IDH1*-*IDH2* Mutated vs. *IDH1*-*IDH2* (Both)WT			
Characteristic	Overall, *n* = 284	*IDH1-IDH2* WT, *n* = 195	*IDH1*-Mut, *n* = 38	*IDH2*-Mut, *n* = 51	*p*-Value ^1^
**Gender, *n* (%)**					0.15
*M*	145 (51%)	93 (48%)	20 (53%)	32 (63%)	
*F*	139 (49%)	102 (52%)	18 (47%)	19 (37%)	
**Age starting treatment,** **median (range)**	65 (19, 86)	65 (19, 85)	66 (22, 86)	65 (32, 85)	0.86
**WBC (10^9^/L), median (range)**	7 (0.5, 800)	8 (0.5, 347)	5 (1, 600)	4 (0.4, 800)	0.063
**HB (g/dL), median (range)**	9.00 (2.50, 14.9)	9.00 (4.50, 14.2)	8.80 (7.20, 13.20)	9.30 (2.50, 14.90)	0.28
**PLTS (10^9^/L), median (range)**	56 (4, 789)	53 (4, 664)	110 (6, 742)	56 (10, 789)	**0.036**
**Blasts (% in BM), median (range)**	50 (3, 99)	48 (4, 99)	75 (3, 96)	70 (4, 95)	**0.027**
**WHO PS, *n* (%)**					**<0.001**
*0*	118 (44%)	79 (43%)	15 (39%)	24 (50%)	
I	111 (41%)	89 (48%)	12 (32%)	10 (21%)	
II	34 (13%)	16 (8.6%)	7 (18%)	11 (23%)	
III	8 (3.0%)	1 (0.5%)	4 (11%)	3 (6.2%)	
**AML type, *n* (%)**					0.63
de novo	229 (81%)	154 (80%)	31 (82%)	44 (86%)	
secondary	37 (13%)	25 (13%)	6 (16%)	6 (12%)	
therapy related	16 (5.7%)	14 (7.3%)	1 (2.6%)	1 (2.0%)	
**AML secondary, *n* (%)**					0.71
*MDS*	24 (65%)	16 (64%)	4 (67%)	4 (67%)	
*ET*	0 (0%)	0 (0%)	0 (0%)	0 (0%)	
*PV*	3 (8.1%)	3 (12%)	0 (0%)	0 (0%)	
*MF*	5 (14%)	2 (8.0%)	1 (17%)	2 (33%)	
***FLT3*, *n* (%)**					0.177
*ITD*	48 (18%)	29 (15%)	13 (38%)	6 (14%)	
*TKD*	10 (3.7%)	9 (4.6%)	1 (2.9%)	0 (0%)	
*ITD and TKD*	2 (0.7%)	2 (1.0%)	0 (0%)	0 (0%)	
***Mutated NPM1*, *n* (%)**	71 (27%)	44 (23%)	17 (50%)	10 (25%)	**0.007**
***Mutated TP53*, *n* (%)**	1 (6.2%)	1 (8.3%)	0 (NA%)	0 (0%)	>0.99
***Mutated CEBPA*, *n* (%)**	3 (9.7%)	2 (15%)	1 (10%)	0 (0%)	0.77
***Mutated IDH1*, *n* (%) **	38 (13%)	0 (0%)	38 (100%)	0 (0%)	**<0.001**
***Mutated IDH2*, *n* (%)**	51 (18%)	0 (0%)	0 (0%)	51 (100%)	**<0.001**
**Complex karyotype, *n* (%)**	29 (11%)	26 (14%)	0 (0%)	3 (6.4%)	**0.021**
**Treatment, *n* (%) **					0.071
*Conventional CHT*	201 (71%)	136 (70%)	29 (76%)	36 (71%)	
*Hypomethylating*	76 (27%)	57 (29%)	8 (21%)	11 (22%)	

^1^ Significant *p*-values are indicated in bold.

**Table 2 cancers-14-03012-t002:** Multivariate models for OS in the whole population of study and in the subset of *IDH1*/*2* mutated patients.

**AML**						
	**Univariate**			**Multivariate**		
**Characteristic**	**HR ^1^**	**95% CI ^1^**	** *p* ** **-Value**	**HR ^1^**	**95% CI ^1^**	** *p* ** **-Value**
**Age**	1.04	1.02, 1.05	**<0.001**	1.03	1.01, 1.05	**0.002**
**WBC**	1.00	1.00, 1.00	**0.004**	1.00	1.00, 1.00	**0.026**
**WHO PS**						
0				–	—	
I	1.96	1.34, 2.88	**<0.001**	1.65	1.09, 2.49	**0.018**
II	2.70	1.63, 4.47	**<0.001**	2.45	1.32, 4.53	**0.005**
III	0.70	0.17, 2.88	0.62	0.54	0.07, 3.99	0.55
**Complex karyotype vs. other**						
other karyotype	-	-		-	-	
complex karyotype	2.83	1.79, 4.45	**<0.001**	3.17	1.91, 5.26	**<0.001**
**Treatment**						
Standard CHT				—	—	
Hypomethylating	2.20	1.56, 3.10	**<0.001**	1.07	0.66, 1.76	0.78
**IDH1/2-mutated AML**						
	**Univariate**			**Multivariate**		
**Characteristic**	**HR ^1^**	**95% CI ^1^**	** *p* ** **-Value**	**HR ^1^**	**95% CI ^1^**	** *p* ** **-Value ^2^**
**Age**	1.03	1.00, 1.06	**0.019**			
**WBC**	1.00	1.00, 1.00	**0.007**	1.00	1.00, 1.01	**0.005**
**Hb**	0.84	0.70, 1.00	**0.049**			
**Treatment**						
Standard CHT				—	—	
Hypomethylating	2.03	1.07, 3.83	**0.03**	1.09	0.48, 2.48	0.84

^1^ HR = hazard ratio; CI = confidence interval; ^2^ Significant *p*-values are indicated in bold.

## Data Availability

Data sharing not applicable.

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
