# Peer review of "Prevalence and Prognostic Role of IDH Mutations in Acute Myeloid Leukemia: Results of the GIMEMA AML1516 Protocol"

_cancers, 2022, doi:10.3390/cancers14123012_

Round 1
Reviewer 1 Report
PREVALENCE AND PROGNOSTIC ROLE OF IDH MUTATIONS IN ACUTE MYELOID LEUKEMIA: RESULTS OF THE GIMEMA AML1516 PROTOCOL
Messina M, PA et al.
Review comments
It is a large cohort of 284 consecutive adult AML focusing on IDH1/2 mutations and clinical outcomes. Per literature, the detected mutation rate of IDH1/2 in AML is 6-17% and 9-19%, respectively. Previous studies showed the two mutations were especially high in those with a normal karyotype (up to 25-30%). Messina M's group demonstrated that IDH1/2 mutations were detected in 12% of AML patients and significantly associated with an inferior performance status (WHO PS), non-complex karyotype and FLT3-WT but not in CR rate and OS at 2 years when compared to IDH1/2-WT. The study is well in written but novelty is concerned. It is recommended to publish in the peer review journal with some modification.
Criteria
1. abstract: It is unclear statement regarding " WBC proved to be independent prognostic factor", high WBCs, or low ANC, or high blast%
2. Please clarify whether GIMEMA AML1516 protocol inclusion criteria. Other adverse risk factors e.g., TP53, inv(3) (ELN risk stratification) have been excluded? The patients with complex cytogenetic abnormalities and harboring MDS related cytogenetic abnormalities (AML-MRC) were included for comparison with those without such aberrations?
3. How many of these AML patients have done completed NGS myeloid mutations (>24 gene panel)? Any correlation with IDH1/2 found?
4. As known, IDH1/2 mutations are associated with NPM1 mutations. Did the co-mutations in your study also override the good prognosis of AML with MPN1?
5. Table 1: blasts% in PB or BM? or calculated circulating blast count
6. Is it possible to analyze the response rate of IDH1/2 mut vs IDH1/2 wt in chemotherapy group vs HMA group? The patients received HMA therapy are mostly secondary AML or AML-MRC with more complex mutational and genetic profile that predict a poor clinical outcome. To compare chemo vs HMA should be in the setting of patients with same disease category. Is IDH1/2 more or less frequently detected in de novo AML than secondary AML?
Author Response
We thank the Reviewer for the comments that give us the possibility to clarify these issues.
- abstract: It is unclear statement regarding " WBC proved to be independent prognostic factor", high WBCs, or low ANC, or high blast%.
High WBC proved to be an indepedent prognostic factor. It is now clarified in the abstract (page 2 line 53) and in the text at page 7 line 221.
- Please clarify whether GIMEMA AML1516 protocol inclusion criteria. Other adverse risk factors e.g., TP53, inv(3) (ELN risk stratification) have been excluded?
Patients were not selected on the basis of any risk factor, neither cytogenetic nor molecular.
The patients with complex cytogenetic abnormalities and harboring MDS related cytogenetic abnormalities (AML-MRC) were included for comparison with those without such aberrations?
Complex karyotype AML- in comparison with the other anomalies – was indeed considered and we found that it was associated with IDH1/IDH2-WT (p=0.016). Similarly, with regards to the AML-MRC, we found that del5q was associated with IDH1/IDH2-WT (p=0.035). This result was added in the revised version of the manuscript at page 6 line 180.
How many of these AML patients have done completed NGS myeloid mutations (>24 gene panel)? Any correlation with IDH1/2 found?
Given the observational nature of the study, the information on the complete NGS myeloid panel was not collected since it was not required for the diagnostic assessment. As stated below, we confirmed the association of IDH1/2 mutations with NPM1 mutations and with FLT3-ITD, that are routinely tested.
- As known, IDH1/2 mutations are associated with NPM1 mutations. Did the co-mutations in your study also override the good prognosis of AML with MPN1?
In the present study, we did not document an impact of IDH1/IDH2 mutations nor on CR achievement neither on survival. Therefore, we did not investigate the impact of IDH-NPM1 co-mutation.
- Table 1: blasts% in PB or BM? or calculated circulating blast count
In Table 1 we reported the percentage of blasts in BM. It is now specified in the table.
- Is it possible to analyze the response rate of IDH1/2 mut vs IDH1/2 wt in chemotherapy group vs HMA group?
Overall, when considering CHT vs HMA, 113 of 181 (62.4%) treated with conventional CHT achieved a CR while only 13 of 41 (31.7%) treated with HMA obtained a CR (p<.0001).
The patients received HMA therapy are mostly secondary AML or AML-MRC with more complex mutational and genetic profile that predict a poor clinical outcome. To compare chemo vs HMA should be in the setting of patients with same disease category.
HMA therapy and AML type were prognostic factors for CR achievement in univariate analysis while complex karyotype did not have an impact. However, HMA therapy did not retain a statistical significance when we adjusted for AML type in the multivariate analysis. Therefore, we can conclude that the benefit of HMA therapy in the same disease category was not observed.
Is IDH1/2 more or less frequently detected in de novo AML than secondary AML?
We did not find a different frequency of IDH1/IDH2 mutations in de novo vs secondary AML. Indeed, 32% of de novo AML carried IDH1/IDH2 mutations and the same percentage was documented in secondary AML.
Reviewer 2 Report
Well written retrospective analysis if IDH 1/2 mutated patients. Thank you for your contribution.
Minor edits/additions:
Please provide the definition of a positive results for IDH mutation based on the two testing methods.
Please note in the methods that IDH 1/2 inhibitors were not used during induction for any of the patients. It should be mentioned.
Comment in the discussion on the use of single agent HMA for IDH-mutated/WT disease. It was inferior to chemotherapy.
Supplemental table: P value of F gender is highlighted (Bold text) but is not statistically significant. Please change.
Author Response
We thank the Reviewer for the comments that give us the opportunity to improve the manuscript.
Minor edits/additions:
Please provide the definition of a positive results for IDH mutation based on the two testing methods.
The sensitivity of Sanger sequencing analysis is approximately 15%-20% and the presence of chromatograms with a double peak into wild-type gene sequence, identified an IDHs gene mutations. For the assessment of IDHs status by NGS assay, the detection limit of the variant allele frequency (VAF) is 5%. These details were added to the revised version of the manuscript (page 4, line 112).
Please note in the methods that IDH 1/2 inhibitors were not used during induction for any of the patients. It should be mentioned.
Given the observational nature of the trial, patients were treated according to local clinical practice that did not entail the administration of IDH1/IDH2 inhibitors. This is now clarified in the text at page 3, line 103.
Comment in the discussion on the use of single agent HMA for IDH-mutated/WT disease. It was inferior to chemotherapy.
As reported in the Result section, the impact of HMA therapy was only observed in univariate analysis. This is the reason why this data was not further mentioned in the Discussion.
Supplemental table: P value of F gender is highlighted (Bold text) but is not statistically significant. Please change.
It was changed.